# Immunizing lithium metal anodes against dendrite growth using protein molecules to achieve high energy batteries

Tianyi Wang[1,5], Yanbin Li[2,5], Jinqiang Zhang [1], Kang Yan[1], Pauline Jaumaux[1], Jian Yang[3], Chengyin Wang[3], Devaraj Shanmukaraj[4], Bing Sun [1✉], Michel Armand [4✉], Yi Cui [2✉] & Guoxiu Wang [1✉]

The practical applications of lithium metal anodes in high-energy-density lithium metal batteries have been hindered by their formation and growth of lithium dendrites. Herein, we discover that certain protein could efficiently prevent and eliminate the growth of wispy lithium dendrites, leading to long cycle life and high Coulombic efficiency of lithium metal anodes. We contend that the protein molecules function as a "self-defense" agent, mitigating the formation of lithium embryos, thus mimicking natural, pathological immunization mechanisms. When added into the electrolyte, protein molecules are automatically adsorbed on the surface of lithium metal anodes, particularly on the tips of lithium buds, through spatial conformation and secondary structure transformation from α-helix to β-sheets. This effectively changes the electric field distribution around the tips of lithium buds and results in homogeneous plating and stripping of lithium metal anodes. Furthermore, we develop a slow sustained-release strategy to overcome the limited dispersibility of protein in the ether-based electrolyte and achieve a remarkably enhanced cycling performance of more than 2000 cycles for lithium metal batteries.

[1] Centre for Clean Energy Technology, University of Technology Sydney, Broadway, Sydney, NSW 2007, Australia. [2] Department of Materials Science and Engineering, Stanford University, Stanford, CA 94305, USA. [3] School of Chemistry and Chemical Engineering, Yangzhou University, 225000 Yangzhou, China. [4] Centre for Cooperative Research on Alternative Energies (CIC energiGUNE), 01510 Vitoria-Gasteiz, Spain. [5]These authors contributed equally: Tianyi Wang, Yanbin Li. ✉email: bing.sun@uts.edu.au; marmand@cicenergigune.com; yicui@stanford.edu; guoxiu.wang@uts.edu.au

Lithium (Li) metal anodes offer the highest theoretical capacity (3860 mAh g$^{-1}$) and lowest electrochemical potential ($-3.04$ V vs. standard hydrogen electrode) among all anode materials for lithium batteries[1]. When coupled with high-capacity cathode materials such as lithium transition metal oxide (LMO), a Li–LMO battery can achieve specific energy densities of 450–500 Wh Kg$^{-1}$, which is double or even triple the capacity of conventional Li–ion batteries[2]. Li metal anodes are indispensable in lithium–sulfur and lithium–oxygen batteries, which can further promote the practical specific energy densities to ~650 Wh Kg$^{-1}$ and ~950 Wh Kg$^{-1}$, respectively[2]. Differing from graphite anodes in Li–ion batteries, Li metal anodes rely on Li stripping and plating, inevitably leading to nucleation and growth of Li dendrites. The growth of Li dendrites causes many severe problems, including low-Coulombic efficiency, short cycle life, short-circuiting, and safety hazards[3].

Recent studies demonstrate that Li deposition in liquid electrolyte involves two different mechanisms[4]. At low current densities and capacities (e.g., under the Sand's capacity), mossy Li grows from the roots. While, above Sand's capacity, wispy Li dendrites quickly grow at the tips. The growth of wispy Li dendrites is a self-amplification process. The 'tip effect' attracts more Li-ions due to the enhanced electric field at the tips of Li dendrites[5]. The hemispherical shape of the tips enables three-dimensional (3D) Li-ion diffusion, rather than the one-dimensional diffusion on a flat surface of Li anode, leading to faster Li plating onto the tips[5]. Therefore, it is essential to inactivate the mossy Li in the initial stage to prevent them from growing into wispy Li dendrites. Many approaches have been developed in attempts to suppress dendrite growth, such as employing biomacromolecule interlayer[6], developing 3D porous current collectors[1,7,8], introducing artificial solid electrolyte interphase (SEI)[9], and optimizing electrolyte formula[10]. Among those strategies, introducing additives in the electrolyte, such as solvents or salts, has been proved to be effective in suppressing Li dendrite formation. So far, there are two types of electrolyte additives reported for Li metal anodes. The first type of additive participates in the formation of SEI, which could significantly enhance the physical property and chemical stability of SEI. The second type of additive absorbs on the tips of the Li protrusions and forms a positively charged electrostatic shield around the tip of the protuberances, which forces further deposition of lithium to adjacent area and suppresses dendrite formation. Despite more than 50 years of research, it remains a significant challenge to achieve 100% dendrite-free Li metal anodes. The ideal solution to completely prevent the formation of wispy Li dendrite would be to automatically inactivate mossy Li as soon as they begin to form, mimicking "early warning" defense responses in biological immunization mechanisms. Nature's biological systems already show evidence of sophisticated multi-step immunization mechanisms. For example, when faced by invading pathogens, antibodies are quickly produced, migrate to the precise locations, and quickly neutralize, inactivate, or destroy the pathogens[11].

In this work, we discover that protein molecules (e.g., fibroin molecules) can effectively prevent lithium metal anodes from severe dendrite growth in lithium metal batteries, resembling natural immunization. We believe this self-defense mechanism enabled by natural protein molecules will undoubtedly bring inspiration for achieving safe and high-energy-density lithium metal batteries.

## Results and discussions

**Self-defense mechanism of fibroin.** Fibroin is selected as a model protein molecule owing to its simple secondary structure for easy characterization[12–15]. It is well recognized that biomolecules such as proteins can be intrinsically adsorbed on the surface of inorganic or metallic materials, and this adsorption process can be further enhanced on the tips or sharp edges of the substrates through electrostatic interactions and cooperative binding effect[16]. The pristine fibroin molecule has a simple helix structure with periodic repeats of three types of amino acids, e.g., alanine residue (ALA), glycine residue (Gly), and serine residue (Ser) [as shown in Fig. 1a (upper-left panel)][17], in which the hydrophobic functional groups such as simple methyl groups (–CH$_3$) are wrapped inside the helix structure and the hydrophilic peptide bonds (–OC–NH–) are exposed externally[18,19]. A visible Tyndall effect confirms that fibroin can be dispersed in the ether-based electrolyte (1M LiTFSI in DOL/DME) in the form of micelles (Supplementary Fig. 1). Furthermore, the fibroin molecules can still be evenly dispersed during the stripping/plating process (Supplementary Fig. 2). When fibroin molecules interact with lithium metal nuclei, we hypothesize that they undergo a conformational change at the secondary structure level from an α-helix to a β-sheet because the β-sheet structure was found to be more thermodynamically stable than an α-helix[20,21]. After the structural and spatial conformational change, the β-sheet fibroin molecules could be easily adsorbed on the tips of mossy Li, reducing the electric field intensity on the tips and preventing the growth of the mossy Li into the wispy Li dendrite. In contrast, lithium dendrites continuously grew when fibroin was not present on the Li metal anode (Fig. 1a).

In order to confirm the above hypothesis, circular dichroic (CD) spectroscopy was employed to analyze the secondary structure transition of protein molecules after cycling. An interlayer of fibroin was prepared by electrospinning and placed on the top of the Li foil in a coin cell. The ether-based electrolyte used in this work is 1 M lithium bis(trifluoromethane sulfonyl) imide (LiTFSI) in a mixed solvent of 1,3-dioxolane (DOL) and 1,2-dimethoxyethane (DME) (1:1 v/v) with 1 wt% lithium nitrate (LiNO$_3$) and 0.5 wt% fibroin as additives. After electrochemical cycling of the coin cell, the interlayer was retrieved from the disassembled coin cell and washed with deionized water. Then, the fibroin solution retrieved from cycled interlayer was analyzed by CD spectroscopy. For comparison, the water solution with pristine fibroin was also prepared for analysis. As shown in Fig. 1b, the CD spectrum of pristine fibroin shows a negative Cotton effect around 198.0 and 202.3 nm, confirming the predominance of the α-helix structure of fibroin molecules[22]. By contrast, we found that the negative Cotton effect moved to 205.8 nm (red curve) in the CD spectrum of cycled fibroin. This unambiguously proves the transformation of protein molecules from α-helices (the result of intramolecular hydrogen bonds) to β-sheets (the result of intermolecular hydrogen bonds)[23]. Such secondary structural transformation of fibroin molecules was further confirmed by Fourier-transform infrared spectroscopy (FT-IR). As shown in Fig. 1c. The broad peak at around 1650 cm$^{-1}$ of the pristine fibroin associated with the amide I band from C=O stretching (~80%) with minor contribution from N–H in-plane blending significantly shifts to 1637 cm$^{-1}$ after cycling. While the original peak at around 1532 cm$^{-1}$ corresponding to amide II from C–N stretching and N–H in-plane blending moves to 1513 cm$^{-1}$. The shift of the signature FT-IR peaks clearly indicates that the secondary structure of fibroin molecules transformed from the original α-helix to β-sheets after cycling[24,25]. To exclude the possibility of secondary structure transformation triggered by Li-ions or organic solvents, a flake of fibroin was immersed in the electrolyte for 1 month and retrieved for characterization. As shown in Supplementary Fig. 3, after being immersed in the electrolyte, two peaks in Amide I area in FT-IR curve of fibroin do not change compared with pristine fibroin. Therefore, it clearly

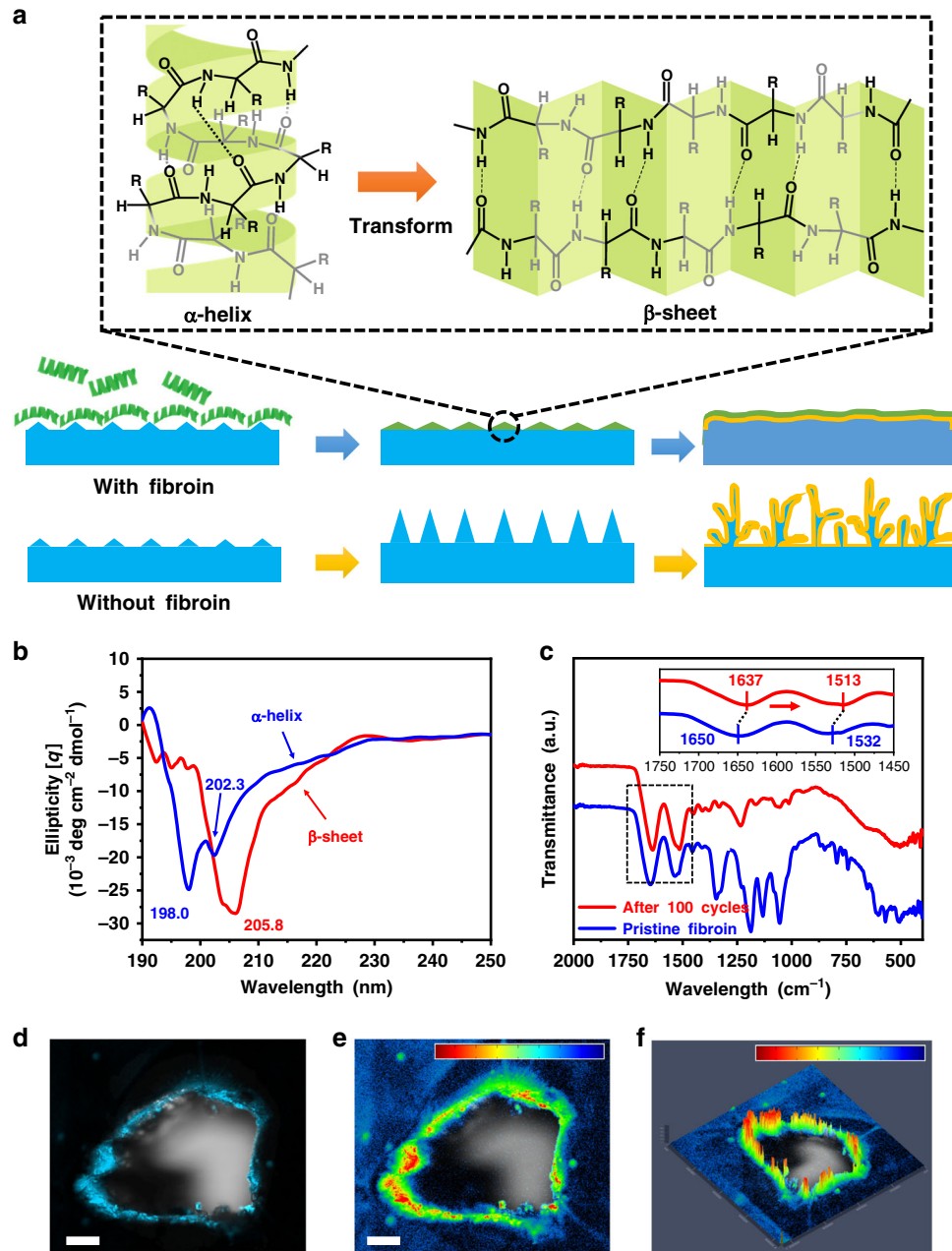

**Fig. 1 Schematic diagram of the self-defense mechanism of fibroin. a** Illustration of the secondary structural transformation of fibroin molecules and the self-defense mechanism of immunizing Li metal anode against Li dendrite growth. ("R" groups are known as side chains of amino acids (-H to Glycine residue, –CH$_2$OH to Serine residue, and –CH$_3$ to Alanine residue)). **b**, **c** CD spectra (**b**) and ATR-FT-IR spectra (**c**) of pristine fibroin and recovered fibroin from the cycled cell after 100 cycles. **d** A fluorescent image of fibroin molecule distribution around the edges and protrusions on a Li metal foil under UV-light. **e**, **f** The corresponding 2D (**e**) and 2.5 D (**f**) simulations of fluorescence intensity. The scale bars on the top in (**e**) and (**f**) correspond to the intensity increase from blue to red. Scale bars, (**d**, **e**) 100 μm.

confirmed that the transformation of the protein secondary structure is not triggered by electrolyte and Li$^+$.

According to the previous report[16], the spatial conformation and high affinity of peptide bonds in fibroin molecules can drive fibroin secondary structure change. When protein molecules with the α-helix secondary structure interact with a Li metal surface, the –NH and C=O functional groups of the peptide bonds show a strong affinity with the Li metal anode, triggering the irreversible conformation change from α-helix (high energy) to β-sheet (low energy). Once the initial layers of fibroin molecules conformably bind to the Li metal surface, the adsorption of further protein molecules can be significantly enhanced owing to cooperative

binding effects[26]. In addition, nanoscale surface roughness such as local protrusions (e.g., Li buds) can also promote the adhesion of protein molecules[27,28]. According to a previous study, fibroin is an insulating material with a high electrical resistivity of 8.5 Ωm[29]. Therefore, the adsorption of fibroin molecules on Li buds can reduce the local electric field intensity and promote the deposition of Li in the surrounding area. We conducted COMSOL™ simulations on the absorption of protein molecules on Li buds, which elucidates the changes in electric field intensity distributions on Li embryos upon the adsorption of protein molecules. As shown in Supplementary Fig. 4a the tip of a Li bud exhibits the highest electric field intensity, promoting further non-uniform

plating and vertical dendritic growth of Li if there is no influence of protein molecules at the close range. When protein molecules are immobilized at the tip of the Li bud, the electric field intensity at the tip rapidly drops from 20 to 10 V m$^{-1}$, while the highest intensity points transfer to the outer border area covered with protein molecules (Supplementary Fig. 4b). When the entire Li bud is fully covered by protein molecules, the growth of Li dendrite from the Li bud will completely stop, leading to Li deposition on the adjacent regions around Li bud until a smooth deposition layer is formed (Supplementary Fig. 4c). A similar strategy has been reported by Zhang et al.[30] to suppress the formation of Na dendrites by changing the electric field distribution through selective Li-ion covering on the tip of Na dendrites.

In order to directly observe the adsorption behavior of protein molecules on the edges and defects of Li metal anodes, we employed the protein fluorescence luminescence method. Fibroin molecules were dyed with fluorescent dye and then dispersed in the ether-based electrolyte. After being immersed in the electrolyte, Li metal electrode was retrieved and washed by DOL to remove Li salts and excess fibroin. To deliberately create defects on the surface of lithium metal foil, we used the tip of a needle to create a pin hole as a controlled defect. As shown in Fig. 1d–f, the adsorbed fibroin molecules emitted clear fluorescence under ultraviolet (UV) light observed by the fluorescence microscope. Especially, fluorescence intensity is much stronger at the edge and sharp tips on Li metal anode, particularly in the region with large curvature. This result corroborates that fibroin molecules prefer to be adsorbed on sharp edges such as dendrites or other defects rather than the flat region, which is consistent with the COMSOL simulation.

**Dense Li deposition with fibroin**. We employed scanning electron microscope (SEM) observation to investigate Li deposition after cycling with and without fibroin additives in the ether-based electrolyte at a current density of 1 mA cm$^{-2}$ with a specific areal capacity limitation of 1 mAh cm$^{-2}$. As shown in Fig. 2a, b, for the cells without fibroin additives in the electrolyte, the top and cross-section views of a Li metal anode with blank electrolyte (i.e., electrolyte with no added fibroin) after 15 cycles exhibited the typical dendritic morphology[31]. The wispy Li dendrites induce a rapid consumption of the electrolyte and short cycle life of the cells. In a sharp contrast, the Li deposition shows a dense and nodule-like morphology when fibroin is added to the electrolyte (Fig. 2c, d)[32]. As shown in Supplementary Fig. 5, even though after 100 cycles, the surface of Li metal anode in the electrolyte with fibroin still maintained a compact surface (Supplementary Fig. 5a, b). While the Li metal anode cycled in the blank electrolyte was covered by coral-like Li dendrites (Supplementary Fig. 5c, d). We further characterized the SEI formed in the electrolyte with fibroin using cryogenic electron microscopy (cryo-EM) (Fig. 2e, f). Li metal was electrochemically deposited on copper grids in a coin cell[33]. Figure 2e shows a typical cryo-EM image of Li nuclei deposited in the electrolyte with fibroin additive. The edge of the Li nuclei with darker contrast corresponds to the SEI region (due to the higher atomic numbers of the SEI components than elemental Li)[31,34]. In a high-magnification cryo-EM image (Fig. 2f), the surface of Li metal shows a bi-layer structure. The outer layer is the absorbed fibroin molecules and the inner layer has been identified as polymeric SEI. According to our previous investigation, the SEI layer formed in blank ether-based electrolyte has a thickness about 17 nm. After we added fibroin molecules in the electrolyte, the thickness of modified SEI has increased from 17 to 28 nm[35,36]. Therefore, this clearly confirmed that the fibroin additive in the electrolyte is involved in the SEI formation.

To in situ monitor the Li deposition behavior, Li | Li symmetric cells are assembled in home-made glass capillaries, as illustrated in Supplementary Fig. 6. In the initial stage, the mossy Li starts to deposit on the Li metal surface in the blank electrolyte. With the increase of the deposition time, the surface of Li metal is fully covered by mossy Li within 20 min. After then, wispy Li dendrites quickly grow on top of the mossy Li deposition layer. In contrast, we did not observe the formation of a large amount of mossy Li on the Li metal anodes in the electrolyte with fibroin under the same deposition condition. The Li deposition layer is compact and homogenous without the formation of the detrimental wispy Li dendrites. Therefore, the addition of fibroin molecules in the electrolyte effectively deactivates mossy Li at the initial stage, preventing them from growing into wispy Li dendrites.

**Characterization of SEI with fibroin**. The physical and chemical properties of the SEI play a critical role in determining the overall electrochemical performance of Li metal anodes. Therefore, X-ray photoelectron spectroscopy (XPS) depth profiling investigations were conducted to examine the composition of SEI formed in the electrolyte with and without fibroin (Fig. 3). The Li | Cu half-cells were disassembled after 10 cycles in the stripped state to analyze the SEI formed on the Cu foil. To avoid the exposure to air, the samples were transferred into the XPS chamber using a sealed argon-filled vessel. In view of the similarity between the XPS spectra of the pristine fibroin (Supplementary Fig. 7) and SEI formed in the electrolytes with and without fibroin (Fig. 3), it is obvious that the SEI formed in the electrolyte with fibroin additive contains fibroin molecule residue. The adsorbed protein molecules on the SEI layer were detected in the N 1$s$ and C 1$s$ spectra. As shown in Fig. 3a, compared with the SEI formed in the blank electrolyte, the N1$s$ spectrum of the SEI formed in the electrolyte containing fibroin shows a new peak at 400 eV (colored in blue). This peak is consistent with the XPS N1$s$ spectrum of the pristine fibroin (Supplementary Fig. 7c). Meanwhile, Fig. 3b also shows that the intensity of the C 1$s$ peak at 288.3 eV (corresponding to C=O) of the SEI formed in the electrolyte with fibroin additive, is higher and larger. Therefore, we suggest that the enhanced C=O peak also comes from the fibroin molecules adsorbed on the surface of the Li metal anode. To investigate how fibroin additive affects the composition of the SEI, Ar$^+$ sputtering was employed to probe different depths of the SEI layer (Supplementary Fig. 8). With the increase of sputtering time, peaks in the N 1$s$ spectra of the SEI layer formed in fibroin-saturated electrolyte are higher than those of the SEI formed in blank electrolyte (Fig. 3a). In particular, an extra peak at 402.8 eV corresponding to α-Li$_3$N appeared in the N 1$s$ spectra. c-Li$_3$N has been widely observed in the SEI layer formed in ether-based electrolytes with LiNO$_3$ additive[33]. Compared with c-Li$_3$N, α-Li$_3$N facilitates the formation of a more mechanically stable SEI with high Li-ion conductivity[37,38]. Furthermore, at different sputtering times, the peak from C=O bond in Fig. 3b was only detected in the SEI formed with fibroin additive. Therefore, fibroin and its decomposed products appear to be involved in the SEI formation. The peaks of O 1$s$ (Fig. 3c) and F 1$s$ (Fig. 3d) are similar for the SEI layers formed in the electrolytes with and without fibroin additive. According to the appearance of Cu 2$p$ spectra (Supplementary Figs. 9 and 10), the thickness of the SEI layer formed in the electrolyte with fibroin is higher than that of the SEI layer formed in the blank electrolyte, which is also consistent with the observation of the cryo-EM analyses.

**Electrochemical performance of Li anodes with fibroin**. We first tested Li | Cu cells using electrolytes with different concentrations of fibroin additive. As shown in Supplementary Fig. 11, after

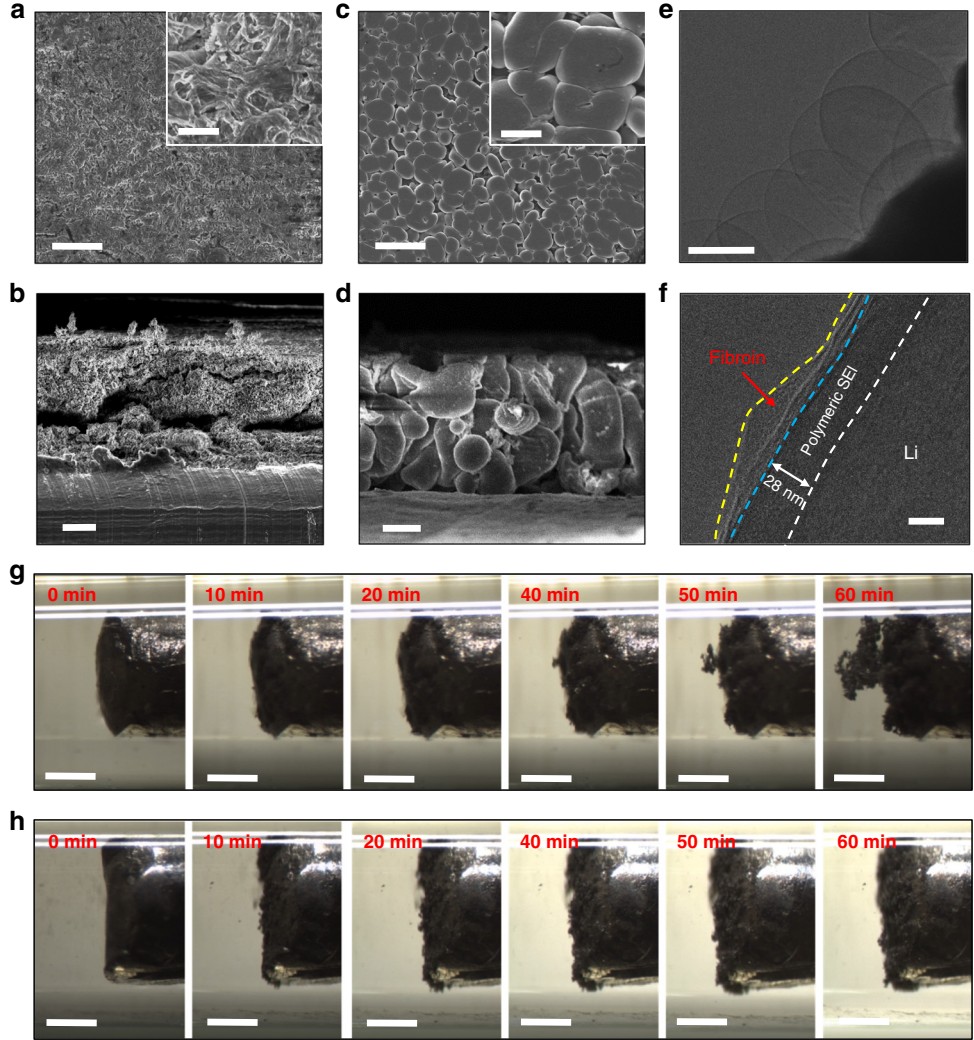

**Fig. 2 Characterization of Li deposition in ether-based electrolyte with fibroin additive. a, b** Top-view (**a**) and cross-section (**b**) SEM images of Li metal anodes cycling in the cells with blank (i.e. no fibroin) electrolyte. **c, d** Top-view (**c**) and cross-section (**d**) SEM images of Li metal anodes cycling in the cells with fibroin additive in the electrolyte. The insets in (**a**) and (**b**) are the corresponding high-magnification SEM images. Scale bars, (**a**–**d**) 20 μm and inserts in (**a**, **c**) 5 μm. **e, f** Low-resolution (**e**) and high-resolution (**f**) cryo-EM images of Li deposit in the ether-based electrolyte with fibroin. Scale bars, (**e**) 1 μm and (**f**) 20 nm. The white line delineates the boundary between metallic Li and SEI. The blue line delineates the boundary between adsorbed fibroin and SEI. The yellow line delineates the boundary between adsorbed fibroin and electrolyte. **g, h** In situ observations of Li deposition behavior in the glass capillaries filled with the blank electrolyte (**g**) and the electrolyte with fibroin (**h**) at a current density of 3 mA cm$^{-2}$. Scale bars, (**g**, **h**) 500 μm.

increase the fibroin concentration from 0.1 to 0.5 wt%, the cycling stability of Li | Cu cells significantly improved. However, when we further increased the concentration of fibroin to 1 wt%, the cycling stability was significantly decreased. The limited dispersibility of fibroin in ether-based electrolyte may be responsible for the degraded cycling performance. When the concentration of fibroin reached to 1 wt%, large amount of fibroin flakes cannot be well dispersed, and many small floccules were observed in the electrolyte. These suspended floccules may unevenly cover the Li metal surface during cell assembling process.

To overcome the dispersibility limitation of fibroin in ether-based electrolyte, we prepared a fibroin interlayer to sustainably release the protein molecules during cycling (Supplementary Fig. 12). The flexible fibroin interlayer consists of nanofibers with abundant micropores, which are beneficial for the absorption of the electrolyte and the release of fibroin (Supplementary Figs. 13 and 14). After immersed in the electrolyte for 3 days, the fibroin interlayer still maintains a good integrity and mechanical property, as shown in Supplementary Fig. 15 and Supplementary

Table 1. The gelling process of fibroin nanofibers in the electrolyte can provide a tighter coverage on the Li metal anodes. Figure 4a shows the voltage profiles of the symmetric Li | Li cells using ether-based electrolytes with and without fibroin interlayers at 3 mA cm$^{-2}$ with the capacity limitation of 1 mAh cm$^{-2}$. The cells with fibroin interlayer exhibited stable voltage profiles over 1000 h. In contrast, the cells without a fibroin interlayer displayed a gradual capacity loss over cycling, and finally failed after 220 h. To further study the evolution of the voltage profiles, those of the symmetric cells at 0–2 h, 200–202 h, 800–802 h, and 1000–1002 h are further enlarged and presented as the inset in Fig. 4a. For the cells with fibroin interlayer, the flat voltage plateau during both plating and stripping remains steady throughout long-term cycling. As the current density increased to 5 mA cm$^{-2}$ (Supplementary Fig. 16), a stable cycling beyond 160 h with stable hysteresis was attained, confirming the exceptional impact of the fibroin interlayer. As a sharp contrast, the cells without the fibroin interlayer exhibited a gradual hysteresis offset during plating/stripping processes. When the capacity limitation was

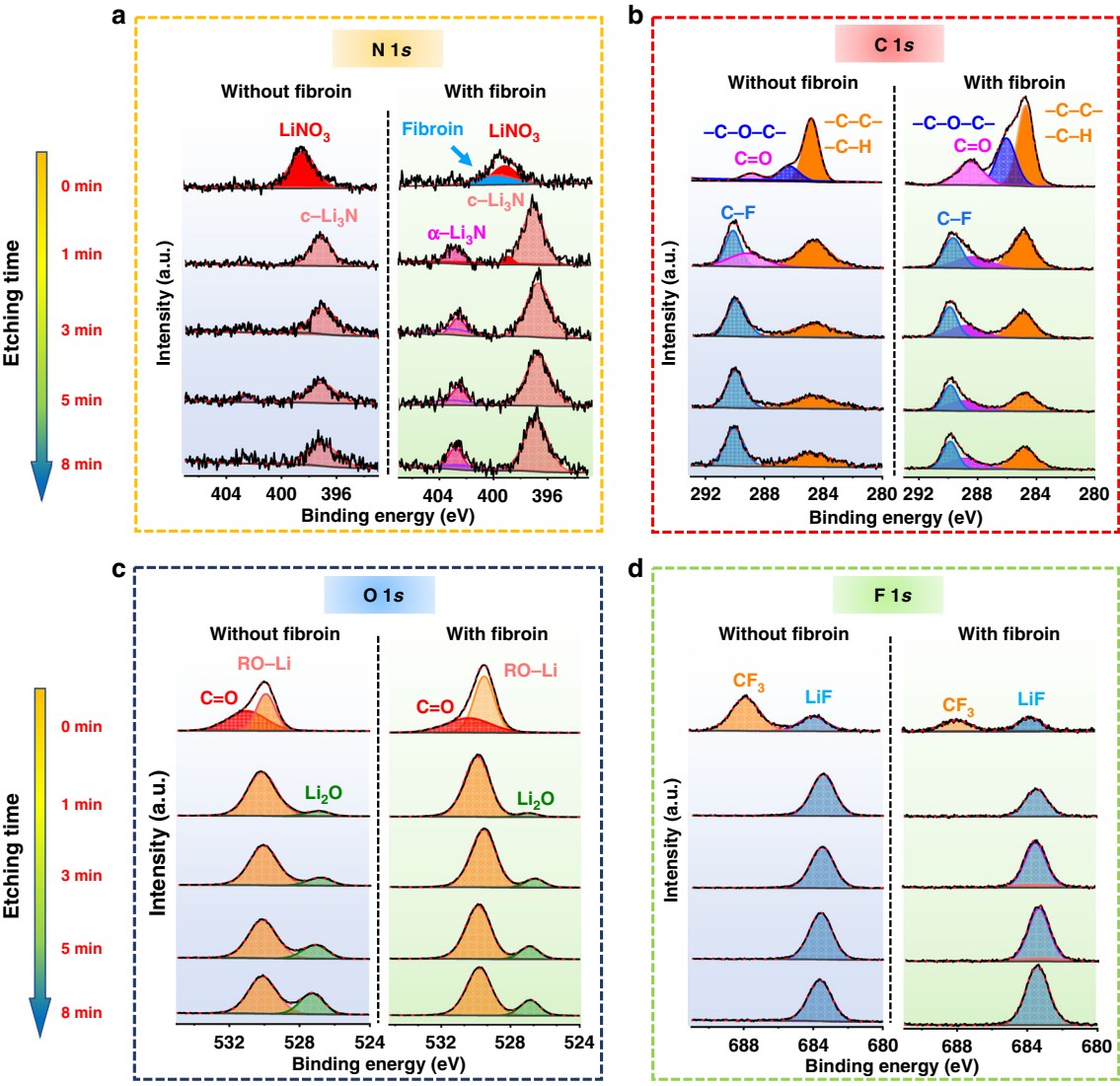

**Fig. 3 X-ray photoelectron spectroscopy (XPS) characterization of SEI formed in ether-based electrolyte with and without fibroin.** Li deposition was carried out at a current density of $1\,mA\,cm^{-2}$ and a capacity of $1\,mAh\,cm^{-2}$ on Cu foil for 10 cycles. **a** N1s spectra, **b** C1s spectra, **c** O1s spectra, and **d** F1s spectra.

increased to $3\,mAh\,cm^{-2}$ and $5\,mAh\,cm^{-2}$, only a slight increase in plating and stripping voltage hysteresis was observed for the cells with fibroin interlayer[39]. However, the cells without fibroin interlayer showed significant voltage hysteresis increases after cycling for 100h (Supplementary Fig. 17).

Coulombic efficiency is an important parameter to evaluate the cycling stability of Li metal anodes and is defined as the ratio between the amount of Li that is stripped from the working electrode and the amount of Li that is plated during each cycle. Considering the cycling life of batteries is related to the electrolyte decomposition upon the reaction with Li metal electrodes, a fair comparison of electrode performance was conducted using a controlled amount of electrolytes (~30 μl of electrolyte per coin cell). To investigate the influence of fibroin interlayer on the Coulombic efficiency of Li metal anodes, Li | Cu half-cells were tested. Li was firstly electrochemically deposited at $1\,mAh\,cm^{-2}$ from the Li metal counter-electrode onto the fibroin interlayer-protected Cu working electrode and then stripped away during the following charge process. Since the Li metal counter-electrode has an excess of Li, the Coulombic efficiency reflects on the Li loss from the Cu working electrode. During cycling, the Li | Cu half-

cells without fibroin failed rapidly due to the generation of Li dendrites, leading to the depletion of the electrolytes cause by the side reaction between the Li metal and the electrolytes. Thus, a gradual decrease in Coulombic efficiency was observed from 95% to lower than 40% after 100 cycles at the current density of 1 mA $cm^{-2}$ (Fig. 4b, c and Supplementary Fig. 18a) and the voltage hysteresis increased upon cycling, with a large overpotential of ~37 mV after 100 cycles (Fig. 4d). As a contrary, the half-cells with fibroin interlayers maintained the Coulombic efficiency as high as ~98% over 100 cycles at a current density of 1 mA $cm^{-2}$ (Fig. 4b, c and Supplementary Fig. 18b). Furthermore, the hysteresis of Li anodes remained stable at ~30 mV for more than 100 cycles (Fig. 4d). To study the stability of the SEI layer, we tested the electrochemical impedance spectroscopy (EIS) of Li | Cu half-cells at the 1st and 50th cycle. As shown in Supplementary Fig. 19, the impedance of the cell with fibroin interlayer is much higher than that of the cell with blank electrolyte in the first cycle. After 50 cycles, the impedance of the cell with blank electrolyte significantly increase. In contrast, the impedance of the cell with fibroin interlayer almost unchanged, indicating significantly improved stability of SEI formed on Li metal anodes[40,41].

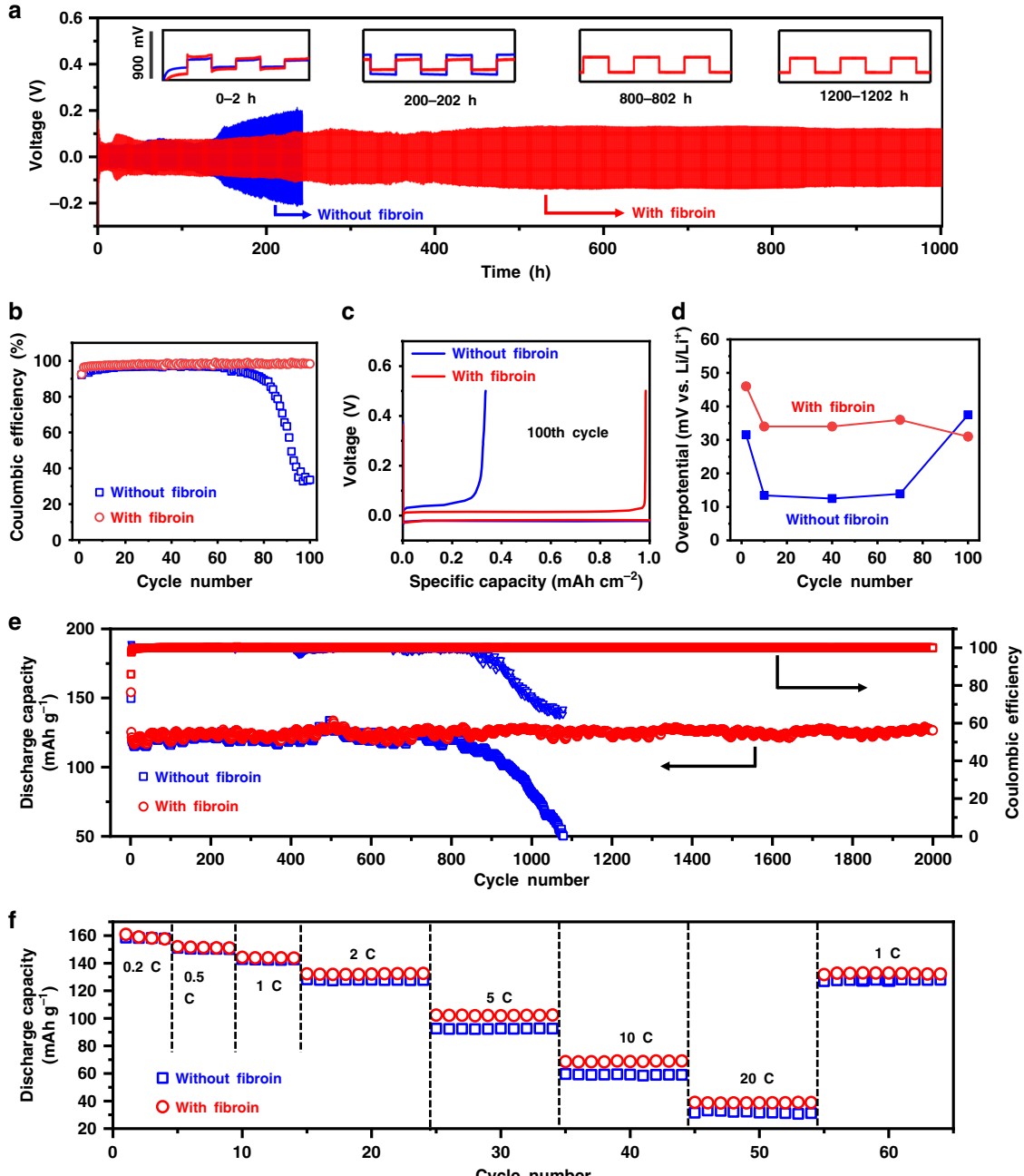

**Fig. 4 The electrochemical performance of Li anode in ether-based electrolyte with or without a fibroin interlayer. a** Galvanostatic cycling of symmetric Li | Li cells with (blue line) or without (red line) a fibroin interlayer. The current density is fixed at 3 mA cm$^{-2}$ with a plating/stripping capacity of 1 mAh cm$^{-2}$. Insets: The blow-up of voltage profiles during 0–2, 200–202, 800–802, and 1000–1002 h, respectively. The y-axis scale of the insets is shown on the left. **b** Comparison of cycling performances of Li | Cu half-cells with or without a fibroin interlayer between Cu foil and separator. The amount of Li deposited in each cycle is 1 mAh cm$^{-2}$ and the current density is 1 mA cm$^{-2}$. **c** The corresponding voltage profiles at the 100th cycle of the Li plating/stripping processes on Cu foil with or without a fibroin interlayer. **d** Comparison of the hysteresis of Li plating/stripping for cells with or without interlayers. **e** Long-term cycling stability of the Li ‖ LTO full cells with or without a fibroin interlayer at a current density of 2 C (1 C = 175 mA g$^{-1}$). **f** Rate capabilities from 0.2 C to 10 C of the Li | LTO cells with or without fibroin additive in the electrolytes.

**Electrochemical performance of Li ‖ Li$_4$Ti$_5$O$_{12}$ full cells.** We further evaluated the electrochemical performances of Li ‖ Li$_4$Ti$_5$O$_{12}$ (LTO) full cells with or without the fibroin interlayer. To exclude the performance deterioration on cathodes, zero-strain LTO was used as the cathode materials. The full cells with fibroin interlayer consistently exhibited better cycling stability and rate capability than those without a fibroin interlayer (Fig. 4e, f). As shown in Fig. 4e, the capacity of the cells with bare Li anodes starts to decay after 800 cycles, indicating the depletion of

active Li and liquid electrolyte. In contrast, the full cells with the fibroin interlayer maintained an excellent capacity over 2000 cycles. This discrepancy is further demonstrated by the voltage vs. capacity profiles of Li ‖ LTO full cells (Supplementary Fig. 20). The voltage polarization of the full cells with bare Li anodes shows an obvious increase after 1000 cycles. However, the voltage polarization of full cells with fibroin interlayer-protected Li anode remains constant for 2000 cycles. Furthermore, as shown in Fig. 4f, the Li ‖ LTO full cells with fibroin interlayer-protected Li

anodes maintain a good high rate capability, with a high-specific capacity of ~102 mAh g$^{-1}$ at 5 C and ~69 mAh g$^{-1}$ at 10 C (xC = fully discharged within 1/x hours). By contrast, Li ‖ LTO cells with blank electrolyte delivered a lower specific capacity of ~93 mAh g$^{-1}$ and ~60 mAh g$^{-1}$ at 5 C and 10 C, respectively.

In conclusion, we have developed an innovative Li anode self-defense strategy inspired by natural immunization mechanisms to eliminate Li dendrite growth and thereby improve the electrochemical performance of Li metal batteries. The natural protein fibroin can effectively prevent Li dendrite nucleation and growth by blocking the evolution of the Li buds in the initial stage. These protein molecules preferably adsorb on Li buds through spatial conformation and secondary structural transformation that significantly affect the local electric field intensity. Therefore, the following Li nuclei will deposit further from the tips of Li buds, leading to a leveling out of Li deposition. Furthermore, to overcome the dispersibility limitation of fibroin in ether-based electrolytes, fibroin interlayers were fabricated to continuously release fibroin in the electrolyte. When applied in a Li‖LTO battery, long-term cycling stability is achieved, and high-specific capacity is delivered even at high rate. The remarkable advantages of this self-defense mechanism enabled by natural protein molecules open up a new and sustainable avenue to achieve safe and dendrite-free high-energy-density lithium metal batteries.

## Methods

**Materials**. The blank ether-based electrolyte is 1 M lithium bis(trifluoromethane sulfonyl)imide (LiTFSI) in a mixed solvent of 1,3-dioxolane (DOL) and 1,2-dimethoxyethane (DME) (1:1 v/v) with 1 wt% lithium nitrate (LiNO$_3$) as additive. The lyophilized silk fibroin flakes with the molecular weight ≥ 1000,000 Daltons were purchased from Simeite company (SF003). To prepare electrolyte with different concentration of fibroin, a certain amount of fibroin is immersed into 10 ml electrolyte. After ultrasonic for 30 min, the electrolyte is kept standstill for 48 h to enable fibroin to fully disperse in the electrolyte. For the preparation of electrospun fibroin interlayer, the fibroin solution is prepared by dissolving the commercial fibroin sponges in deionized water under stirring for 1 h. The concentration of fibroin solution for electrospinning is controlled to be 12 wt%. The electrospun fibroin interlayer is collected on a target rotating drum, which is placed opposite to the syringe tip with a distance of 15 cm. A voltage of 25 kV is applied to the collecting target by a high voltage power supply (Chungpa EMT Co., Korea).

**Characterization**. Circular dichroism (CD) spectra are collected using a Jasco H-810 spectropolarimeter equipped with a NESLAB RTE-111 slab and purged with N$_2$ gas at a flow rate of 3 to 5 ml min$^{-1}$. The fibroin aqueous solution was spotted in 0.10 cm path length cells for detection. CD spectra were recorded from 190 to 250 nm wavelength with a resolution of 0.2 nm and an accumulation of five scans at a scanning rate of 100 nm min$^{-1}$ and the response time of 0.25 s. A blank solution was measured under the same experimental conditions and the blank levels were subtracted from the data. The ratio of α-helix and β-sheet structure was analyzed by the protein secondary structure estimation software (Protein secondary structure estimation). The Fourier-transform infrared spectroscopy (FT-IR) was used as an effective method to measure fibroin. The adsorption features were recorded using a Nicolet 6700 FT-IR spectrometer in the range of 1800–1200 cm$^{-1}$ and the resolution of 4 cm$^{-1}$. The fibroin interlayer was retrieved from the Li | Li symmetric cell. After cycling for 50 cycles at a current density of 5 mA cm$^{-2}$, the fibroin interlayer was removed from coin cell in the Ar-filled glove box and washed by 1 ml DOL to remove the salt. Before FT-IR characterization, the fibroin interlayer was dried under vacuum for 12 h. To process the protein fluorescence luminescence measurement, the fibroin was firstly dyed by 8-Anilino-1-naphthalenelsfonic acid and dispersed in the electrolyte. After immersed in the electrolyte for 2 h, the Li metal electrode was taken out and washed by DOL solvent for 3 times. To prevent being oxidized in air, the Li foil was sealed in a quartz dish in Ar. The two-photon excitation microscopy (Cari Zeiss, LSM 88-NLO) was employed to monitor the fluorescence phenomenon on Li metal electrode under UV-light. Distribution of fluorescence intensity was analyzed by ZEN 3.2 software. X-ray photoelectron spectroscopy (XPS) measurements were carried out using a ThermoFisher Scientific ESCALAB250Xi with a monochromatic Al Ka X-ray source (3000 eV) at 150 W and 15 kV with a beam spot size of 500 μm. The samples were transferred into the vacuum chamber using a sealed Ar-filled vessel. Depth profiling was conducted using Ar ion sputtering with an accelerate voltage of 0.5 kV over a 2.5 × 2.5 mm area to 0.5 μm depth. The thickness of the SEI was estimated from the calibrated sputtering rate of 0.3 nm per second in Ta$_2$O$_5$ and the sputtering time at which the atomic concentration of Li drops to <5% (by

measurement or extrapolation). The Cu 2$p_{3/2}$ peaks arise due to the underlying of Cu foil. Field-emission scanning electron microscopy (FE-SEM, Zeiss Supra 55VP) was used to investigate the morphologies of the as-prepared fibroin interlayer and the cycled electrodes.

**Electrochemical measurements**. For the Li | Li symmetric cell assembling, coin cells (CR2032) are assembled in an Ar-filled glove box (MBraun, H$_2$O < 0.1 p.p.m, O$_2$ < 0.1 p.p.m), using two pieces of Li metal disks, two pieces of fibroin interlayers, ether-based electrolyte, and Celgard$^{TM}$ 2325 separator. The ether-based electrolyte is 1 M LiTFSI in DOL/DME (1:1 v/v) containing 1 wt% LiNO$_3$ with or without fibroin. For Li | Cu half-cell assembling, coin cells were assembled using Li foil as the counter-electrode and Cu foil as the working electrode. For the Coulombic efficiency testing, in each galvanostatic cycle, Li was deposited on Cu foil at the desired current density and capacity, and stripped away by charging to a cutoff voltage of 1.0 V vs. Li$^+$/Li. The Li plating/stripping study was conducted on Neware battery testers at room temperature. Electrochemical impedances were measured using a CHI660E electrochemical station with a frequency range of 0.1 Hz to 100 kHz. For Li ‖ Li$_4$Ti$_5$O$_{12}$ (LTO) full cell test, LTO electrodes are prepared first. The weight ratio of LTO (BTR Co.) material, carbon black (CB) and sodium carbonxymethly cellulose (CMC) in the LTO electrode is 8:1:1. The Li ‖ LTO full cells were assembled using Li metal electrode, fibroin interlayer, 30 μL electrolytes, Celgard$^{TM}$ 2325 separator, and LTO cathode electrode. All cells were assembled in an Ar-filled glove box (H$_2$O < 0.1 ppm, O$_2$ < 0.1 ppm). Li ‖ LTO full cells were first activated at 0.2 C for 5 cycles within a voltage window between 1.0 and 2.5 V. Then, they were cycled at 2 C for long-term cycling stability test.

## Data availability
The data that support the finding of this study are available from the corresponding author upon request.

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

## Acknowledgements

This project is financially supported by the Australian Research Council (ARC) through the ARC Discovery projects (DP170100436, DP180102297 and DP200101249) and ARC Discovery Early Career Researcher Award (DE180100036).

## Author contributions

G.W., B.S., and T.W. conceived the idea. Y.L. performed the cryo-EM characterization. Y.C. participated in the design of experiment and supported technical guidance. T.W., J.Z, B.S., K.Y., and P.J. carried out the electrochemical experiments. T.W. and B.S. performed characterization of Li metal anodes. C.W. and J.Y. participated in the experiment of fluorescence probing of fibroin. M.A. and D.S. supported technical guidance and writing of the manuscript.

## Competing interests

The authors declare no competing interests.
