## [Peer Review File · Nature Communications]

Reviewers' comments:

Reviewer #1 (Remarks to the Author):

The authors prepared a fibroin separator for lithium metal batteries, and demonstrated that the electrodeposits of lithium become nodule-like smooth. The reviewer is highly impressed by the inspiration the authors took from the natural systems. However, based on the scientific concerns below, the current manuscript should not be accepted for publication.

1. Li structures grown in liquid system follow a root-growing mechanism, as shown in a paper by Bai et al., *Energy & Environmental Science*, 2016. Due to this dynamic process, it's hard to justify the "tip effects" once considered in earlier papers. Simply put, the "dendrites" (which should be whiskers) do not grow from the tip. The authors may therefore reconsider the mechanisms they proposed.

2. I was not able to find any mathematical details for the COMSOL simulation shown in Fig. 1. What are the equations, parameters, boundary conditions? Simulation results cannot justify experiments, unless these choices I mentioned are all justified physically.

3. In figure 2 panel C and D, what was the cycle number for these deposits? Since the authors showed a perfect cycling efficiency even after 2000 cycles. The impact of this work would be maximized if they can include do a morphological comparison with deposits after 2000 cycles. Presumably, they should be as smooth.

4. The authors are highly encouraged to read the paper by Albertus published in *Nature Energy*, <https://www.nature.com/articles/s41560-017-0047-2>, the community is focusing on lean Li metal electrode and cycling at areal capacity higher than 3 mAh cm⁻². The authors should try this higher areal capacity to see if the good performance can survive.

5. The current work look quite similar to ref 5. The authors should provide a detailed comparison to justify their innovation.

Reviewer #2 (Remarks to the Author):

In this manuscript, Tianyi Wang et al. studied the electrochemical properties of fibroin protein as an additive for the ether-based electrolyte (1 M LiTFSI in DOL-DME) for advanced dendrite-free lithium metal battery. Author added fibroin protein to the electrolyte and confirmed its structure through XPS depth profiling, cryo-EM, and high resolution transmission electron microscopy and presented functional results via electrochemistry. The followings are my comments and suggestions.

1. What is the mechanism of the interaction between Li buds and this selected protein? Simply providing CD spectrum and XPS data can't give convincing explanation. More evidences should be provided.

2. The authors argue that the transformation of protein molecules from α -helices to β -sheets owing to intramolecular hydrogen bonds, why is it not the denaturation of proteins caused by some metal ions (Li⁺) or organic electrolyte that leads to the breaking of peptide bonds?

3. How to confirm that protein molecules are evenly dispersed during the stripping/plating states instead of aggregating at the electrolyte?

4. In Figure 2F, there is no obvious interface between Li metal and polymeric SEI, is there any more sufficient evidence about the formation and composition of the SEI layer?

5. Author claimed that the fibroin protein molecules are automatically adsorbed on the surface of lithium metal structure. In Figure 2F, whether such thick SEI (28 nm) can block the intertation of Li metals to silk fibroin protein?

6. In Figure 3A, how to distinguish the N1s spectrum of the SEI formed in the electrolyte

containing fibron, which shows a new peak at 400 eV (colored in blue) rather than N1s spectrum of the LiTFSI?

7. The Li || LTO full cells were assembled using Li metal electrode ($\Phi=16$ mm), 30 μ L electrolytes, CelgardTM 2325 separator ($\Phi=18$ mm), fibron interlayer and LTO cathode electrode. How thick is the silk fibron protein layer? In addition, what is the effect on the volume energy density of the full cell?

8. What are the effects of different molecular weight and concentration of fibron protein on lithium metal battery? More discussions are required.

9. In electrolyte the protein is in organic-gel state, how is its mechanical property and dimensional integrity?

Reviewer #3 (Remarks to the Author):

Wang et al. reported the use of protein molecules for reviving lithium metal batteries. Although the results are interesting, the manuscript needs to be revised heavily both from the scientific and editorial perspective. Detail analysis and further characterizations are needed at different conditions (additives including different concentrations and interlayer including the different thickness). Therefore, I think the paper can be considered for publication in nature communications after addressing the following major comments.

1. The top-view and cross-sectional SEM images while using fibron as an additive in the electrolyte was studied. The detailed SEM analysis while using the fibron as an interlayer also needs to be studied. The optimization of fibron concentration for the additive based and the thickness optimization for the interlayer based could be done. The dispersibility limitation of fibron was not discussed in detail.

2. What is the thickness of fibron interlayer? The thickness of SEI film with fibron participation is higher than that of the SEI film formed in the blank electrolyte. What is the real thickness of only SEI while using fibron interlayer and without using fibron interlayer? Further stability of the SEI can be studied by EIS measurement before and after cycling with or without fibron modified lithium. The controlled thickness of Li deposition with artificial SEI and the stability in the impedance measurement (EIS) more clearly explains the stability and robustness of SEI. Please cite and discuss Nature Communications 11.1 (2020): 1-10 and Advanced Energy Materials, 9(36), 1901486 for more details.

3. The results of the transformation of α -helix to β -sheet are demonstrated. How does the fibron molecule interact with lithium metal nuclei? The physics behind this transformation needs to be further discussed.

4. The fibron molecules are insulating type. After the transformation of α -helix to β -sheet still, remain insulating type? During plating/stripping cycles, the lithium deposition is underneath the interlayer or within the structure or on the top?

5. After electrochemical cycling of the coin cell, the interlayer was retrieved from the disassembled coin cell and washed with deionized water. Washing with deionized water doesn't change the morphology or structure. For washing and removing the unnecessary residues from the electrode, before further characterizations, the DME or DOL are common solvents.

6. How the authors can claim that peptide bonds in β -sheet fibron molecules are lithiophilic. Further discussion is needed.

7. The use of LTO cathode can compromise the voltage to achieve high energy density batteries. The high capacity/high voltage cathodes in carbonate electrolyte can be used for testing the suitability of the fibron modified lithium.

8. The in-situ formation of stable SEI consumes both lithium and electrolyte. How this work addresses this issue needs to be discussed in the introduction. If the author could add a discussion about the advantages of this approach compared to common methods, it will be persuasive to readers. A discussion in the background will help readers to understand the manuscript.

Minor comments:

1. The active mass of the LTO cathode electrode is missing.

2. The thickness of fibron interlayer and its mass loading is missing in the experimental section. The use of thicker interlayer with high additional mass could also compromise the energy density of the battery.

3. The nucleation overpotential for a symmetric cell with fibron showed higher nucleation overpotential compared to without fibron in Fig. 4A (inset 0-2 h) in the beginning hours. In general, the lithiophilic coating lowers the nucleation overpotential in the beginning cycles and also the overpotential in higher plating/stripping cycles. The detailed discussion is missing.

Response to reviewers:

Reviewer 1

The authors prepared a Fibroin separator for lithium metal batteries and demonstrated that the electrodeposits of lithium become nodule-like smooth. The reviewer is highly impressed by the inspiration the authors took from the natural systems. However, based on the scientific concerns below, the current manuscript should not be accepted for publication.

Response: We thank to the reviewer's comments and have done a series of experimental investigations to elucidate the mechanism of fibroin for the protection of Li metal anodes through suppressing the formation of detrimental Li dendrites. These methods include *in situ* observation of Li electrodeposition in a transparent glass cell, fluorescent image of fibroin molecule distribution around the controlled protrusions and sharp edges on the Li foil under UV-light, more SEM observations on cycled Li metal anodes, and mechanical property testing of fibroin interlayer, etc.

Question 1: *Li structures grown in liquid system follow a root-growing mechanism, as shown in a paper by Bai et al., Energy & Environmental Science, 2016. Due to this dynamic process, it's hard to justify the "tip effects" once considered in earlier papers. Simply put, the "dendrites" (which should be whiskers) do not grow from the tip. The authors may therefore reconsider the mechanisms they proposed.*

Response: We agree with the reviewer's comments and changed the corresponding description in the introduction part in the revised manuscript (Please see Page 2 in the revised manuscript). Actually, the opinions raised by the reviewer do not contradict with the mechanism proposed in this work. In the previous publication mentioned by the reviewer (Bai et al. *Energy Environ. Sci.*, **2016**, 9, 3221), the growth mechanism of Li dendrite during Li ion deposition can be classified into two stages. In the first stage, mossy Li mainly grows from its root, which is below the limiting current and attributed to internal stress release beneath the SEI layer on the lithium metal anodes. In the second stage, wispy Li dendrites grow explosively from their tips and easily penetrate the separator to cause short-circuit.

In our research, we mainly focus on deactivating mossy Li at the initial stage (e.g., the "first stage" mentioned in the above-mentioned EES article), preventing them from developing into wispy Li

dendrites. To provide more evidence, we assembled Li | Li symmetric cells in the glass capillaries to support the discussion of our current proposed mechanism, which has been added in the revised manuscript as shown below:

“To *in situ* monitor the Li deposition behavior, Li | Li symmetric cells are assembled in home-made glass capillaries, as illustrated in **Fig. S6**. In the initial stage, the mossy Li starts to deposit on the Li metal surface in the blank electrolyte. With the increase of the deposition time, the surface of Li metal is fully covered by mossy Li within 20 min. After then, wispy Li dendrites quickly grow on top of the mossy Li deposition layer. In contrast, we do not observe the formation of a large amount of mossy Li on the Li metal anodes in the electrolyte with fibroin under the same deposition condition. The Li deposition layer is compact and homogenous without the formation of the detrimental wispy Li dendrites. Therefore, the addition of fibroin molecules in the electrolyte effectively deactivates mossy Li at the initial stage, preventing them from growing into wispy Li dendrites.” (Please see Page 6 Line 27 in the revised manuscript)

Fig. 2: *In situ* observations of Li deposition behavior in the glass capillaries filled with (G) blank electrolyte and (H) electrolyte with fibroin additive at a current density of 3 mA cm^{-2} . (Please see Page 7 in the revised manuscript)

The corresponding experiment description has been added in the revised Supporting Information.

“Transparent Li | Li symmetric glass cell: The middle necklet of glass capillary was pulled thinner. A small amount of electrolyte was filled in by the capillary effect. Then, two small pieces of Li metal flaks were pushed into two terminals of the capillary. The silicon glue was used to seal the glass tube. Cu wires were used as the current collector.” (Please see Page 4 Line 18 in Supplementary information)

The configuration of the glass battery cell is shown below:

Fig. S6. Schematic illustration of the configuration of Li | Li symmetric glass cell assembled in a homemade glass capillary. (Please see Page 11 in Supplementary information)

Question 2: I was not able to find any mathematical details for the COMSOL simulation shown in Fig. 1. What are the equations, parameters, boundary conditions? Simulation results cannot justify experiments, unless these choices I mentioned are all justified physically.

Response: Thanks for the reviewer’s comments. The simulation details are presented below and have been added in the Supporting Information.

“The stationary equation of continuity is used to handle the stationary electric currents in conductive media. In a stationary coordinate system, the point form of Ohm’s law states:

$$\mathbf{J} = \sigma \mathbf{E} + \mathbf{J}_e$$

Where σ is the electrical conductivity (S m^{-1}), and \mathbf{J}_e is an externally generated current density (A m^{-2}). The static form of the equation of continuity then states:

$$\nabla \cdot \mathbf{J} = -\nabla \cdot (\sigma \nabla V - \mathbf{J}_e) = 0$$

To handle the current sources, we generalize the equation to:

$$-\nabla \cdot (\sigma \nabla V - \mathbf{J}_e) = \mathbf{Q}_j$$

In planar two-dimension stimulation, the Electric Currents interface assumes that the model has a symmetry, where the electric potential varies only in the x and y directions and is constant in the z direction. This implies that the electric field, \mathbf{E} , is tangential to the xy -plane. The Electric Currents interface then follows the following equation, where d is the thickness in the z direction:

$$-\nabla \cdot d(\sigma \nabla V - \mathbf{J}_e) = d\mathbf{Q}_j \text{ (Please see Page 4-5 in Supplementary information)}$$

***Question 3:** In Figure 2 panel C and D, what was the cycle number for these deposits? Since the authors showed a perfect cycling efficiency even after 2000 cycles. The impact of this work would be maximized if they can include do a morphological comparison with deposits after 2000 cycles. Presumably, they should be as smooth.*

Response: Thanks for the reviewer's comments. The cycle number in Figure 2 C and D is 15 cycles, which is the same as in Fig. 2A and B (Please see page 5, Line 5 in the manuscript). Currently, many methods have been investigated to suppress the formation of Li dendrites, and some of them have achieved outstanding cycle life in Li metal-based batteries [1-5]. However, the morphologies of Li metal anodes still changed significantly after long-term cycling. The claim of "suppressing dendrite formation" in most of the previous studies does not mean that the surface can be kept as smooth as its original status. After cycling, the surface of Li metal anodes became rough but without the formation of obvious wispy Li dendrite before the failure of the cells.

In our current work, the morphologies of Li metal anodes changed as well after cycling. In the blank electrolyte (i.e., electrolyte without added fibroin) after 15 cycles, the Li metal anode exhibited a typical dendritic morphology with large amount of mossy Li dendrites formed on the surface. In contrast, the Li metal anode cycled in the electrolyte with fibroin showed nodule-like morphology. The morphology difference of the cycled Li metal anodes between two electrolytes already demonstrated the essential effect of fibroin for achieving dendrite-free Li metal anodes.

To further investigate the morphology changes after long-term cycling, we also added the SEM images

of Li metal surfaces after 100 cycles in the Supporting Information.

“As shown in Fig. S5, even though after 100 cycles, the surface of Li metal anode in the electrolyte with fibroin still maintained a compact surface (Fig. S5A and B); While the Li metal anode cycled in the blank electrolyte was covered by coral-like Li dendrites (Fig. S5C and D).” (Please see Page 6 Line 12 in the revised manuscript)

Fig. S5. SEM images of Li metal surface cycled in the electrolyte with fibroin after 100 cycles (A, B) and in the blank electrolyte after 100 cycles (C, D). The current density is 1 mA cm^{-2} under a stripping/plating capacity of 1 mA h cm^{-2} . (Please see Page 10 in Supplementary information)

References:

1. Li, G.X., et al., *Organosulfide-plasticized solid-electrolyte interphase layer enables stable lithium metal anodes for long-cycle lithium-sulfur batteries*. Nature Communications, 2017. **8** article number 850.
2. Liu, Y.Y., et al., *Lithium-coated polymeric matrix as a minimum volume-change and dendrite-free lithium metal anode*. Nature Communications, 2016. **7**, article number 10992.
3. Yang, C.P., et al., *Accommodating lithium into 3D current collectors with a submicron skeleton towards long-life lithium metal anodes*. Nature Communications, 2015. **6**, article number 8058.
4. Liang, X., et al., *A facile surface chemistry route to a stabilized lithium metal anode*. Nature Energy, 2017. **2**(9), article number 17119.
5. Lin, D.C., et al., *Layered reduced graphene oxide with nanoscale interlayer gaps as a stable host for lithium metal anodes*. Nature Nanotechnology, 2016. **11**(7), 626-632.

Question 4: The authors are highly encouraged to read the paper by Albertus published in Nature Energy, <https://www.nature.com/articles/s41560-017-0047-2>, the community is focusing on lean Li metal electrode and cycling at areal capacity higher than 3 mAh cm⁻². The authors should try this higher areal capacity to see if the good performance can survive.

Response: Thanks for the reviewer's suggestion. Actually, the electrochemical performances at higher areal capacity above 3 mAh cm⁻² have been presented in our originally submitted manuscript (**Please see Page 10, Line 26 in the revised manuscript**). The voltage vs. time curves of Li || Li symmetrical cells cycled with the capacity limitations of 3 mAh cm⁻² and 5 mAh cm⁻² have been shown in Figure S10 in the Supplementary Information. In addition, we have cited the above-mentioned reference as Ref. 39 in the revised manuscript.

Question 5: *The current work looks quite similar to ref 5. The authors should provide a detailed comparison to justify their innovation.*

Response: Thanks for the reviewer's valuable suggestion. There are significant differences between our work and Ref.5. In the work of Ref.5, Ju *et al.* employed the natural membrane from the eggshell to control lithium dendrite growth behavior. The eggshell membrane was modified by TFEA to improve their lithiophilicity. When used as an interlayer between separator and Li metal anode, the eggshell membrane plays the role of a Li-ion re-distributor, which can reduce the growth of lithium dendrites at the initial stage. However, that research did not provide a systematic elucidation about the mechanism for the protection of Li metal anodes. In particular, that work did not identify which kind of protein is useful because the eggshell membrane consists of many different proteins and other bio species.

In our research, we selected commercial fibroin as an example and proposed a novel mechanism. According to our investigations, fibroin can be dispersed in ether-based electrolytes and maintain their natural properties (secondary structure can be maintained in the electrolyte). During the Li deposition

process, protein molecules are adsorbed on the tip of mossy Li and preventing it from growing into wispy Li dendrite through altering the distribution of the electric field. Upon immobilized on the tip of moss Li, the protein molecules were de-natured with their secondary structure transformation from α -helix to β -sheet. This transformation is self-induced, which can enhance the interaction between fibroin and moss Li.

Reviewer 2

In this manuscript, Tianyi Wang et al. studied the electrochemical properties of fibroin protein as an additive for the ether-based electrolyte (1 M LiTFSI in DOL-DME) for advanced dendrite-free lithium metal battery. Author added fibroin protein to the electrolyte and confirmed its structure through XPS depth profiling, cryo-EM, and high-resolution transmission electron microscopy and presented functional results via electrochemistry. The followings are my comments and suggestions.

Response: We thank the reviewer to give us an opportunity to revise this manuscript. All the reviewer's comments have been replied point-by-point. Some additional experimental results have been added in the revised manuscript and revised supporting information.

Question 1: *What is the mechanism of the interaction between Li buds and this selected protein? Simply providing CD spectrum and XPS data can't give convincing explanation. More evidence should be provided.*

Response: Thanks for the reviewer's comment. It is well recognized that biomolecules such as proteins can be intrinsically adsorbed on the surface of inorganic or metallic materials, and this adsorption process could be further enhanced on the tips or the sharp edges of the substrates (Please refer to Reference 15, *Chem. Rev.* 2011, **111**, 5610). When fibroin molecules interact with Li metal anodes, they undergo a conformational change at the secondary structure level from an α -helix to a β -sheet because the β -sheet structure is more thermodynamically stable than an α -helix. After the structural and spatial conformational change, the β -sheet fibroin molecules could be easily adsorbed on the tips of mossy Li, reducing the electric field intensity on the tips and preventing the growth of the mossy Li into the wispy Li dendrite.

To further explore the interactions between Li buds and fibroin molecules, we use protein fluorescence luminescence method to demonstrate protein adsorption behavior on the sharp edges and protrusions of Li metal anodes. The following description and figures are added in the revised manuscript.

“In order to directly observe the adsorption behavior of protein molecules on the edges and protrusions of Li metal anodes, we employed the protein fluorescence luminescence method. Fibroin molecules were dyed with fluorescent dye and then dispersed in the ether-based electrolyte. After being immersed in the electrolyte for 1h, Li metal electrode was retrieved and washed by DOL to remove Li salts and excess fibroin. To deliberately create protrusions and tips on the surface of lithium metal foil, we used a needle to make a pin hole on the Li foil to create sharp edges, tips and protrusions. As shown in Fig. 1D-F, the adsorbed fibroin molecules emitted clear fluorescence under ultraviolet (UV) light observed by the fluorescence microscope. Especially, fluorescence intensity is much stronger at the edge and sharp tips on Li metal anode, particularly in the region with large curvature. This result corroborates that fibroin molecules prefer to be adsorbed on sharp edges such as dendrites or other protrusions rather than in the flat region, which is consistent with the COMSOL simulation.” (Please see Page 4, Line 28 in the revised manuscript)

Fig. 1 (D) A fluorescent image of fibroin molecule distribution around the edges and protrusions on a lithium metal foil under UV-light. The corresponding (E) 2D and (F) 2.5 D simulations of fluorescence intensity. The scale bars on the top in E and F correspond to the intensity increase from blue to red. (Please see Page 5 in the revised manuscript)

Question 2: *The authors argue that the transformation of protein molecules from α -helices to β -sheets owing to intramolecular hydrogen bonds, why is it not the denaturation of proteins caused by some*

metal ions (Li^+) or organic electrolyte that leads to the breaking of peptide bonds?

Response: Thanks for the reviewer's comment. To explicitly explain the mechanism of the transformation of protein molecules from α -helices to β -sheets, we investigated the fibroin secondary structure in the ether-based electrolyte. The corresponding description has been added in the revised manuscript.

“To exclude the possibility of secondary structure transformation triggered by Li ions or organic electrolyte, a flake of fibroin was immersed in the electrolyte for 1 month and retrieved for characterization. As shown in Fig. S3, after being immersed in the electrolyte, two peaks in Amide I area in FT-IR curve of fibroin do not change compared with pristine fibroin. Therefore, it clearly confirmed that the transformation of the protein secondary structure is not triggered by electrolyte and Li^+ .” (Please see Page 3 Line 37 in the revised manuscript).

In addition, Figure S3 has been added in the revised supporting information.

Figure S3. Comparison of FT-IR curves of fibroin before and after immersed in the electrolyte for 1 month. (Please see Page 8, in the supplementary information)

Question 3: How to confirm that protein molecules are evenly dispersed during the stripping/plating states instead of aggregating at the electrolyte?

Response: Thanks for the reviewer's comment. To confirm the even distribution of protein molecules during stripping/plating, we assembled Li | Li symmetric cell in a glass vessel to observe Tyndall effect. As shown in **Fig. S2**, we can still observe Tyndall effect during stripping/plating process at a current density of 1 mA cm^{-2} , which confirms that fibroin molecules are dispersed homogeneously in the electrolyte during cycling. The corresponding description has been added in the revised manuscript.

“Furthermore, the fibroin molecules can still be evenly dispersed during the stripping/plating process (Fig. S2).” (Please see Page 3 Line 10 in the revised manuscript).

Fig. S2. Tyndall effect of fibroin dispersed electrolyte during Li | Li plating/stripping process. We can still observe Tyndall effect in a transparent Li | Li symmetric cell during stripping/plating process at a current density of 1 mA cm^{-2} , which confirms that fibroin molecules are dispersed homogeneously in the electrolyte during cycling. (Please see Page 7 in Supplementary Information)

Question 4: In Figure 2F, there is no obvious interface between Li metal and polymeric SEI, is there any more sufficient evidence about the formation and composition of the SEI layer?

Response: Thanks for the reviewer's comments. The enlarged Cryo-EM image is shown below, which shows the interface of the SEI layer. In addition, the composition of the SEI layer formed in the electrolyte with and without fibroin additive were characterized by XPS in-depth profiles (Fig. 3 and Fig. S8-10). The XPS results show fibroin and its decomposed products appear to be involved in the SEI formation.

Figure R1. Enlarged Cryo-EM image.

Question 5: Author claimed that the fibroin protein molecules are automatically adsorbed on the surface of lithium metal structure. In Figure 2F, whether such thick SEI (28 nm) can block the interaction of Li metals to silk fibroin protein?

Response: Thanks for the reviewer's comments. The SEI could block the direct interactions between Li metal and fibroin. However, if the Li dendrites protrude from the SEI, fibroin molecules could be adsorbed on the tips of mossy Li to suppress their further growth.

Question 6: In Figure 3A, how to distinguish the N1s spectrum of the SEI formed in the electrolyte containing fibroin, which shows a new peak at 400 eV (colored in blue) rather than N1s spectrum of the LiTFSI?

Response: Thanks for the reviewer's comment. Before the XPS measurement, all samples have been washed by 2 ml DOL twice to remove the residence Li salts. From Fig. 3A, we did not observe any peaks related to LiTFSI from the Li metal anodes cycled in the blank electrolyte. Therefore, all LiTFSI and LiNO₃ should have been removed except the immobilized fibroin molecules on the surface of Li metal anodes. From the N1s XPS spectra of pristine fibroin in Fig. S7c, we also observe a strong peak at 400 eV.

***Question 7:** The Li || LTO full cells were assembled using Li metal electrode ($\Phi=16$ mm), 30 μ L electrolytes, CelgardTM 2325 separator ($\Phi=18$ mm), fibroin interlayer and LTO cathode electrode. How thick is the silk fibroin protein layer? In addition, what is the effect on the volume energy density of the full cell?*

Response: Thanks for the reviewer's comment. We measured the thickness of the fibroin interlayer. The thickness of the fibroin interlayer is about 20 μ m, which is similar to the commercial separator used in this work (about 23 μ m). The influence of the fibroin interlayer to the volume energy density of batteries is almost negligible. We use this interlayer to release fibroin molecules sustainably and homogeneously into the electrolyte.

***Question 8:** What are the effects of different molecular weight and concentration of fibroin protein on lithium metal battery? More discussions are required.*

Response: The fibroin used in this work is a commercial product with the molecular weight of more than 200,000 Dal. Other fibroin products with different molecular weights are not commercially available. However, we believe the molecular weight of the fibroin may influence their dispersibility in the electrolyte. Nevertheless, we prepared 3 electrolytes with different concentrations of fibroin. The following testing results and discussion have been added in the revised manuscript and revised supporting information.

“We first tested Li | Cu cells using electrolytes with different concentrations of fibroin additive. As shown in Fig. S11, after increase the fibroin concentration from 0.1 wt% to 0.5 wt%, the cycling stability of Li || Cu cells significantly improved. However, when we further increased the concentration

of fibroin to 1 wt%, the cycling stability was significantly decreased. The limited dispersibility of fibroin in ether-based electrolyte may be responsible for the degraded cycling performance. When the concentration of fibroin reached to 1 wt%, large amount of fibroin flakes cannot be well dispersed, and many small floccules were observed in the electrolyte. These suspended floccules may unevenly deposit on the Li metal surface during cell assembling process.” (Please see Page 10 Line 1 in the revised manuscript)

Fig. S11. Comparison of Coulombic efficiency of Li || Cu half cells using electrolyte with different fibroin concentrations. (Please see Page 16 in Supplementary Information).

Question 9: In electrolyte the protein is in organic-gel state, how is its mechanical property and dimensional integrity?

Response: Thanks for the reviewer’s suggestion, we have tested the mechanical properties of the fibroin interlayer before and after immersed in the electrolyte for 3 days. The following results have been added in the revised manuscript and revised supporting information

“After immersed in the electrolyte for 3 days, the fibroin interlayer still maintains a good integrity and mechanical property, as shown in Fig. S15 and Table S1. The gelling process of fibroin nanofibers in the electrolyte can provide a tighter coverage on the Li metal anodes.” (Please see Page 10 Line 12 in the revised manuscript, and Page 20 and Page 21 in Supplementary Information)

Fig. S15 Digital photos of (A) a pristine fibroin interlayer and (b) a fibroin interlayer retrieved from electrolyte after immersed for 3 days.

Table S1. Comparison of mechanical properties of fibroin interlayer before and after immersed in ether-based electrolyte for 3 days.

Fibroin interlayer	Tensile strength (Mpa)	Elongation at break (%)	Young's modulus (Mpa)
Dry electrolyte	8.1±2.4	18.1±2.5	167±35
After immersed in electrolyte for 3 days	12.9±3.1	24.2±1.7	115±12

Reviewer #3 (Remarks to the Author):

Wang et al. reported the use of protein molecules for reviving lithium metal batteries. Although the results are interesting, the manuscript needs to be revised heavily both from the scientific and editorial perspective. Detail analysis and further characterizations are needed at different conditions (additives including different concentrations and interlayer including the different thickness). Therefore, I think the paper can be considered for publication in nature communications after addressing the following major comments.

Response: We thank the reviewer to give us an opportunity to revise this manuscript. All the concerns have been replied one by one with additional data added in the revised manuscript to support our conclusion.

***Question 1:** The top-view and cross-sectional SEM images while using fibroin as an additive in the electrolyte was studied. The detailed SEM analysis while using the fibroin as an interlayer also needs to be studied. The optimization of fibroin concentration for the additive based and the thickness optimization for the interlayer based could be done. The dispersibility limitation of fibroin was not discussed in detail.*

Response: Thanks for the reviewer's reminding. We have added the SEM image of Li metal anode while using fibroin interlayer in Li | Li symmetry cell. When fibroin interlayer was employed between separator and Li metal electrode, the morphology changes of Li metal anode is similar to the result shown in Fig. 2C after 15 cycles.

Fig. R1. SEM image of Li metal anode after 15 cycles using fibroin interlayer in Li | Li symmetry

battery cell.

To optimize the concentration of fibroin additive in the electrolyte, we prepared 3 electrolytes with different concentrations of fibroin. The following testing results and discussion have been added in the revised manuscript.

“We first tested Li | Cu cells using electrolytes with different concentrations of fibroin additive. As shown in Fig. S11, after increase the fibroin concentration from 0.1 wt% to 0.5 wt%, the cycling stability of Li || Cu cells significantly improved. However, when we further increased the concentration of fibroin to 1 wt%, the cycling stability was significantly decreased. The limited dispersibility of fibroin in ether-based electrolyte may be responsible for the degraded cycling performance. When the concentration of fibroin reached to 1 wt%, large amount of fibroin flakes cannot be well dispersed, and many small floccules were observed in the electrolyte. These suspended floccules may unevenly deposit on the Li metal surface during cell assembling process.” (Please see Page 10 Line 1 in the revised manuscript)

Fig. S11. Comparison of Coulombic efficiency of Li || Cu half cells using electrolyte with different fibroin concentrations. (Please see Page 16 in Supplementary Information).

Question 2: *What is the thickness of fibroin interlayer? The thickness of SEI film with fibroin participation is higher than that of the SEI film formed in the blank electrolyte. What is the real thickness of only SEI while using fibrin interlayer and without using fibroin interlayer? Further stability of the SEI can be studied by EIS measurement before and after cycling with or without fibrin modified lithium. The controlled thickness of Li deposition with artificial SEI and the stability in the impedance measurement (EIS) more clearly explains the stability and robustness of SEI. Please cite and discuss Nature Communications 11.1 (2020): 1-10 and Advanced Energy Materials, 9(36), 1901486 for more details.*

Response: We thanks the reviewer for this comment and have cited the recommended references in the revised manuscript as Ref. 39 and 40. The thickness of the fibroin interlayer is about 20 μm . In this work, fibroin interlayer is used to release fibroin molecules sustainably and homogeneously into the electrolyte. Therefore, the fibroin interlayer should be as thin as possible if they can form an integrated film. The thickness of the SEI on Li metal anodes should be similar as long as there are fibroin molecules in the electrolytes.

In addition, the EIS measurement has been conducted on the Li | Cu half cells before and after cycling with or without fibroin interlayers. The results below were added in the Supporting Information.

“To study the stability of the SEI layer, we tested the electrochemical impedance spectroscopy (EIS) of Li | Cu half cells at the 1st and 50th cycle. As shown in **Fig. S19**, the impedance of the cell with fibroin interlayer is much higher than that of the cell without the interlayer in the first cycle. After 50 cycles, the impedance of the cell with blank electrolyte significantly increase. In contrast, the impedance of the cell with fibroin interlayer almost unchanged, indicating significantly improved stability of SEI formed on Li metal anodes. [40, 41]” (Please see Page 11 Line 8 in the revised manuscript).

Figure S19. The EIS measurement of Li | Cu cells at the 1st and 50th cycles (A) without and (B) with the fibroin interlayer at the current density of 1 mA cm^{-2} with the capacity limitation of 1 mAh cm^{-2} . (Please see Page 25 in Supplementary Information)

Reference added in the revised manuscript:

[40] R. Pathak, K. Chen, A. Gurung, K. Reza, B. Bahrami, F. Wu, A. Chaydhary, N. Ghimire, B. Zhou, W. Zhang, Q. Qiao, Ultrathin Bilayer of Graphite/SiO₂ as solid interface for Reviving Li metal anode, *Adv. Mater.* **2019**, 9, 1901486-1901495.

[41] R. Pathak, K. Chen, A. Gurung, K. Reza, B. Bahrami, J. Pokharel, A. Banuya, W. He, F. Wu, Y. Zhou, K. Xu, Q. Qiao, Fluorinated hybrid solid-electrolyte-interphase for dendrite-free lithium deposition, *Nat. Commun.* 2020, 11, 93.

Question 3: The results of the transformation of α -helix to β -sheet are demonstrated. How does the fibroin molecule interact with lithium metal nuclei? The physics behind this transformation needs to be further discussed.

Response: Thanks for the reviewer's comment. It is well recognized that biomolecules such as proteins can be intrinsically adsorbed on the surface of inorganic or metallic materials, and this adsorption process could be further enhanced on the tips or the edges. When fibroin molecules interact with Li metal anodes, they undergo a conformational change at the secondary structure level from an α -helix

to a β -sheet because the β -sheet structure is more thermodynamically stable than an α -helix. After the structural and spatial conformational change, the β -sheet fibroin molecules could be easily adsorbed on the tips of mossy Li, reducing the electric field intensity on the tips and preventing the growth of Li into a dendrite morphology.

To further explore the interaction between Li buds and fibroin, we use protein fluorescence luminescence method to demonstrate protein adsorption behavior on the edges and defects of Li metal anodes. The following description and figures are added in the revised manuscript.

“In order to directly observe the adsorption behavior of protein molecules on the edges and protrusions of Li metal anodes, we employed the protein fluorescence luminescence method. Fibroin molecules were dyed with fluorescent dye and then dispersed in the ether-based electrolyte. After being immersed in the electrolyte for 1h, Li metal electrode was retrieved and washed by DOL to remove Li salts and excess fibroin. To deliberately create protrusions and tips on the surface of lithium metal foil, we used a needle to make a pin hole on the Li foil to create sharp edges, tips and protrusions. As shown in Fig. 1D-F, the adsorbed fibroin molecules emitted clear fluorescence under ultraviolet (UV) light observed by the fluorescence microscope. Especially, fluorescence intensity is much stronger at the edge and sharp tips on Li metal anode, particularly in the region with large curvature. This result corroborates that fibroin molecules prefer to be adsorbed on sharp edges such as dendrites or other protrusions rather than in the flat region, which is consistent with the COMSOL simulation.” (Please see Page 4, Line 28 in the revised manuscript)

Fig. 1 (D) A fluorescent image of fibroin molecule distribution around the edges and protrusions on a lithium metal foil under UV-light. The corresponding (E) 2D and (F) 2.5 D simulations of fluorescence

intensity. The scale bars on the top in E and F correspond to the intensity increase from blue to red.
(Please see Page 5 in the revised manuscript)

Question 4: *The fibroin molecules are insulating type. After the transformation of α -helix to β -sheet still, remain insulating type? During plating/stripping cycles, the lithium deposition is underneath the interlayer or within the structure or on the top?*

Response: Thanks for the reviewer's comments. Both α -helix and β -sheet fibroin molecules are electronic insulate. We disassembled a coin cell after deposited 1 mAh cm⁻² of Li on Cu foil. As shown in Figure R2, after exfoliated the fibroin interlayer, we can observe a layer of Li deposited on Cu current collector. Meanwhile, there is no resident Li observed on top of fibroin interlayer.

Fig. R2 Digital photo of a disassembled coin cell with fibroin interlayer.

Question 5: *After electrochemical cycling of the coin cell, the interlayer was retrieved from the disassembled coin cell and washed with deionized water. Washing with deionized water doesn't change the morphology or structure. For washing and removing the unnecessary residues from the electrode, before further characterizations, the DME or DOL are common solvents.*

Response: Thanks to the reviewer's reminding. The fibroin interlayer was washed by DOL first after retrieved from the cycled cell. For further CD characterization, the fibroin needs to be dissolved in aqueous solution because the organic solvation effect could interfere the results of DC spectrum.[1, 2]

References:

[1] Y. Yang, Z. Shao, X. Chen, P. Zhou, Optical Spectroscopy to Investigate the Structure of Regenerated Bombyx mori Silk Fibroin in solution, *Biomacromolecules*, 2004, 5, 773-779.

[2] M. Canetti, A. Seves, F. Secundo, G. Vecchio, CD and small-angle x-ray scattering of silk fibroin in solution, *Biopolymers*, 1989, 28, 1613-1624.

***Question 6:** How the authors can claim that peptide bonds in β -sheet fibroin molecules are lithiophilic. Further discussion is needed.*

Response: Thanks to the reviewer's comment and we realize the use of "lithiophilic" here is not appropriate. We have deleted this description in the revised manuscript.

***Question 7:** The use of LTO cathode can compromise the voltage to achieve high energy density batteries. The high capacity/high voltage cathodes in carbonate electrolyte can be used for testing the suitability of the fibroin modified lithium.*

Response: Thanks to the reviewer's suggestion. The fibroin has very low dispersibility in carbonate-based electrolytes. Therefore, we only focus on the ether-based electrolyte in this work.

***Question 8:** The in-situ formation of stable SEI consumes both lithium and electrolyte. How this work addresses this issue needs to be discussed in the introduction. If the author could add a discussion about the advantages of this approach compared to common methods, it will be persuasive to readers. A discussion in the background will help readers to understand the manuscript.*

Response: Thanks to the reviewer's suggestion, we have added the discussion about the advantages of this approach compared to other methods in the introduction part in the revised manuscript.

"Among those strategies, introducing additives in the electrolyte, such as solvents or salts, has been

proved to be effective in suppressing Li dendrite formation. So far, there are two types of electrolyte additives reported for Li metal anodes. The first type of additive participates in the formation of SEI, which could significantly enhance the physical property and chemical stability of SEI. The second type of additive adsorbs on the tips of the Li protrusions and forms a positively charged electrostatic shield around the tip of the protuberances, which forces further deposition of lithium to adjacent area and suppresses dendrite formation.” (Please see Page 2 Line 15 in the revised manuscript)

“The fibroin molecules work as a multifunctional additive in the electrolyte. They participate in the formation of a stable SEI, which effectively blocks the parasitic reaction between Li metal and electrolyte. Meanwhile, the adsorption of fibroin molecules on the tips of mossy Li reduces the electric field intensity on the tips and prevents the growth of the mossy Li into the wispy Li dendrite.” (Please see Page 2 Line 33 in the revised manuscript)

Minor comments:

1. The active mass of the LTO cathode electrode is missing.

Response: The mass loading of LTO electrode for full cell testing is about 5.4 mg cm^{-2} .

2. The thickness of fibroin interlayer and its mass loading is missing in the experimental section. The use of thicker interlayer with high additional mass could also compromise the energy density of the battery.

Response: We measured the thickness of the interlayer. The thickness of the fibroin interlayer is about $20 \text{ }\mu\text{m}$, which is similar to the commercial separator used in this work (about $23 \text{ }\mu\text{m}$). The addition of this fibroin interlayer will slightly reduce the energy density of the cells. We use this interlayer to release fibroin molecules sustainably and homogeneously into the electrolyte. Therefore, the fibroin interlayer should be as thin as possible if they can form an integrated film. Future investigation will be explored to optimize the synthesis process to further reduce the thickness of this interlayer.

3. The nucleation overpotential for a symmetric cell with fibroin showed higher nucleation overpotential compared to without fibroin in Fig. 4A (inset 0-2 h) in the beginning hours. In general,

the lithiophilic coating lowers the nucleation overpotential in the beginning cycles and also the overpotential in higher plating/stripping cycles. The detailed discussion is missing.

Response: The authors thank the reviewer for the valuable comment. Protein coating on the surface of Li metal anodes is not lithiophilic coating. Due to the electronic insulation property, the adsorption of fibroin molecules could reduce the electric field intensity on the tips of mossy Li, preventing it from developing into wispy Li dendrite. The corresponding description has been revised in the manuscript.

REVIEWERS' COMMENTS

Reviewer #1 (Remarks to the Author):

I appreciate the serious revisions made by the authors.

Fig 2 needs scale bars for the capillary cells.

Reviewer #2 (Remarks to the Author):

The authors have well answered all questions raised by the reviewers in the new version. The supplemented data are convincing and their hypothesis can be generally proven.

Not necessarily, but I still feel query about this work as follows:

1. Is it a general method for introducing proteins into lithium metal batteries? Why choosing silk fibroin?
2. From SI fig. S13, the electrospun fibroin film appears to show a rough surface, which suggests that the distribution of fibroin chains on Li surface could be inhomogeneous. What if directly adding fibroin powders into the electrolyte solution as a gel?
3. The underlying mechanism of the transformation from α -helix to β -sheet can be briefly discussed.

Reviewer #3 (Remarks to the Author):

The authors have thoroughly revised the manuscript according to the comments from reviewers. I am satisfied with the changes and would recommend accepting this work for publication in Nature Communications.

One minor change is to correct the journal of reference 40 from Adv Mater. to "Advanced Energy Materials 9 (36), 1901486, 2019".

Response to reviewers:**Reviewer 1****Comments to author:**

I appreciate the serious revisions made by the authors. Fig 2 needs scale bars for the capillary cells.

Response: Thanks for the reviewer's comments. We have added the scale bars in Figure 2 for the capillary cells.

Reviewer 2**Comments to author:**

The authors have well answered all questions raised by the reviewers in the new version. The supplemented data are convincing and their hypothesis can be generally proven.

Not necessarily, but I still feel query about this work as follows:

Question 1: *Is it a general method for introducing proteins into lithium metal batteries? Why choosing silk fibroin?*

Response: Thanks for the reviewer's comments. One of the general methods for introducing proteins into lithium metal batteries is to disperse them into the selected electrolyte. Fibroin has been widely employed in many applications such as the textile industry and the pharmaceutical industry. The major reason that we chose fibroin as a model protein molecule is its simple secondary structure for easy characterization. Meanwhile, fibroin is inexpensive and has been produced as a commercial product in large scale.

Question 2: *From SI fig. S13, the electrospun fibroin film appears to show a rough surface, which suggests that the distribution of fibroin chains on Li surface could be inhomogeneous. What if directly adding fibroin powders into the electrolyte solution as a gel?*

Response: Thanks for the reviewer's suggestion. The dispersibility of fibroin is limited in ether-based electrolyte. When the concentration of fibroin reached to 1 wt%, large amount of fibroin flakes cannot be well dispersed, and many small floccules were observed in the electrolyte. These suspended floccules may unevenly cover the Li metal surface during cell assembling process. Therefore, to

overcome the dispersibility limitation of fibroin in ether-based electrolyte, we prepared a fibroin interlayer to sustainably release the protein molecules during cycling

Question 3: *The underlying mechanism of the transformation from α -helix to β -sheet can be briefly discussed.*

Response: Thanks for the reviewer's suggestion. The transformation from α -helix to β -sheet may follow two mechanisms. The first mechanism involves direct transition of the random coil part of the helical conformation into antiparallel β -sheet. The second mechanism is related to that the α -helical conformation unfolded and converted into antiparallel β -sheet.

Reviewer 3

Comments to author:

The authors have thoroughly revised the manuscript according to the comments from reviewers. I am satisfied with the changes and would recommend accepting this work for publication in Nature Communications. One minor change is to correct the journal of reference 40 from Adv Mater. to "Advanced Energy Materials 9 (36), 1901486, 2019".

Response: Thanks for the reviewer's reminding. We have corrected the reference in the revised manuscript.

Reference cited in the revised manuscript:

40. Pathak R., Chen K., Gurung A., Reza K., Bahrami B., Wu F., Chaydhary A., Ghimire N., Zhou B., Zhang W., Qiao Q., Ultrathin Bilayer of Graphite/SiO₂ as solid interface for Reviving Li metal anode, *Adv. Energy Mater.* **9**, 1901486-1901495 (2019).